# What integrated care means from an older person's perspective? A scoping review protocol

Manasi Murthy Mittinty, Amy Marshall, Gillian Harvey

Adelaide Nursing School, The University of Adelaide, Adelaide, Australia

**Correspondence to**
Dr Manasi Murthy Mittinty; manasi.mittinty@adelaide.edu.au

## ABSTRACT

**Introduction**  According to the 2013 WHO Global Forum on Innovation for Ageing Populations, disabilities and morbidities associated with ageing could be minimised by accessing preventive care. One way of improving the management of multimorbidity in the older population is through the provision of 'integrated care'. Although integrated care means different things to different people, it typically symbolises continuity in care, thus preventing older patients' from falling through gaps in the health care system. Many initiatives have attempted to improve the integration of care; however, these are typically designed from a particular policy or system perspective. Relatively little is known about patient expectations and experiences of integrated care, which is vital for developing and implementing better models of care. The proposed scoping review aims to map literature on older patients'' views, expectations, experiences and perspectives of integrated care.

**Methods and analysis**  Multiple electronic databases including PubMed, Web of Science, Embase, PsychInfo, Google Scholar, Cochrane Library, CINAHL and ProQuest Dissertations will be searched for appropriate articles between August and December 2017. Reference lists of selected articles will also be searched for similar articles. Two experienced researchers will conduct an initial search of the literature to identify relevant articles. Abstracts of the identified articles will be reviewed collectively by two researchers to identify potential further studies. Full texts of the potential studies will be sourced and screened for the inclusion criteria. Appropriate qualitative and quantitative methods will be used to extract data from each included study.

**Ethics and dissemination**  The scoping review will synthesise findings from studies reporting on patients' views and expectations of integrated care. This review expects to find information relating to facilitators and barriers of integrated care from an older person's perspective. The findings from the review will be applied when working with stakeholders representing older people, healthcare, aged care and community providers, researchers and policy makers to develop and evaluate a more locally tailored and person-centred approach to integrated care.

## Strengths and limitations of this study

► This study will be the first scoping review to provide a comprehensive synthesis of what older people consider to be integrated care, what does it mean to them and what do they expect.

► This study will search all sources of literature covering peer-reviewed articles, unpublished reports, conference proceedings and bibliographies.

► Stakeholders involved in the provision of or affected by integrated care, including older people, healthcare providers, government organisations, carers and family members, will be engaged throughout the review process.

► Scoping reviews are generally not considered to provide generalisable findings because of the lack of synthesis of results; conducting a thematic analysis of the available literature will go some way to providing further insight into the findings than descriptive data alone.

► As this review will incorporate studies from different healthcare settings, it may lack specificity.

a resultant increase in the proportion of older people in the population. This shift in demographics has both benefits and challenges. Older people are a great social and cultural resource; they are also lower financial contributors and higher healthcare system users. As older people often suffer from multimorbidity, they typically receive treatments from more than one health professional, consume multiple medications and have multiple organisations and service providers involved in their care.[3] As a result, older people tend to experience more fragmented care, which can lead to more preventable acute hospitalisations,[4–6] placing additional strain on patients', their families, carers and healthcare systems. In turn, this contributes to suboptimal health outcomes, reduced quality of life for patients' and increased healthcare costs.[7 8]

Integrated care is proposed as a solution to both improve patient care and minimise the unnecessary use of healthcare resources.[9] However, it represents a major challenge

## BACKGROUND

In the last 40 years, life expectancy has increased significantly on a global level[1 2] with

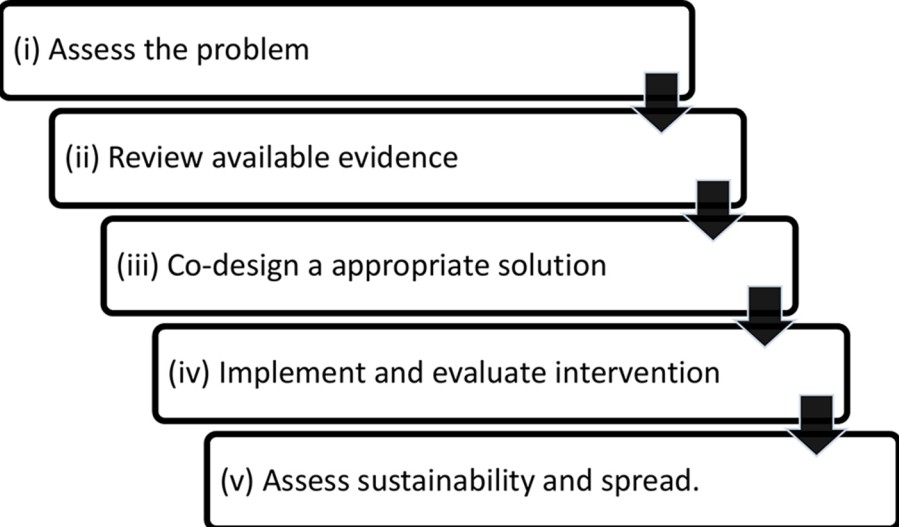

**Figure 1** Co-creation knowledge translation framework.

facing health service providers and policy makers.[10] Despite numerous attempts at integration, uncertainty remains about which approaches are most effective[9] or how to achieve an appropriate form of integration at a local level. Empirical evidence highlights the complexities of achieving integrated care within pluralistic delivery systems with multiple stakeholders, varying cultures and different mechanisms of funding and governance. While it is clear that there are no 'one-size-fits-all' solutions, successful approaches are typically bottom-up, driven by local need and with the support and engagement of all key stakeholders, including patients' and their family.[11]

However, despite the growing body of scientific evidence relating to integrated care and its implications, few studies[12 13] have summarised patients' perspectives and expectations of integrated care. There is a lack of comprehensive understanding of what older people themselves consider to be integrated care, what does it mean to them and what do they expect. Some research suggests that there may be distinct differences between medical and patient narratives, with patients' emphasising the importance of relational aspects of care and the everyday consequences of their condition, for example, in terms of functional impairment and feelings of vulnerability,[14] important factors that influence a person's ability to manage and coordinate their own care.[13 15] This reinforces the importance of understanding patients' perspectives and views of integrated care. The proposed scoping review aims to map literature on patients' views, expectations, experiences and perspectives of integrated care. This paper describes the protocol that will be followed for conducting the scoping review.

### Study rationale

Globally, a large number of heterogeneous, disease-specific and setting-specific models of integrated care are being implemented. However, it remains unclear the extent to which these models have been validated, in terms of meeting a person-centred view of integrated care. The necessity to incorporate patients' views when building these models has been identified by several studies.[16–18] Currently, there is a paucity of information on what factors patients' perceive to support or hinder their care. To our knowledge, only one published study has synthesised patient experiences and expectations of integrated care.[19] In the absence of a clear person-centred definition of integrated care, the scoping review seeks to facilitate the development of such a definition. It will also provide insight into this poorly understood area of research and help to map the characteristics of the primary research that currently exists, as well as identifying knowledge gaps.

The objective of the scoping review is to systematically examine the range and scope of literature and to determine enablers and barriers of integrated care from an older person's perspective. The review forms part of a larger programme of work to coproduce and evaluate locally relevant approaches to improve integrated care for older people at risk of repeated hospitalisation. This programme is informed by a conceptual framework known as the *co-creation knowledge translation framework*[20] (co-KT framework) (figure 1). The framework was previously developed and tested in a population health study in a regional area of South Australia.[21] The basic tenet of the framework is that solutions to health service problems are best tackled by working with affected groups or communities to clarify and assess the problem and collectively develop, implement and evaluate appropriate evidence-informed solutions. Our pilot work looking at the coordination and continuity of care for older people in a defined geographical area of South Australia has highlighted opportunities for improving the integration of care between acute, primary and community care providers.[22] Combining this local knowledge with the co-KT framework, the scoping review forms part of our evidence gathering strategy to define the characteristics of integrated care from the perspective of those affected,

**Table 1** Selection criteria to be used for identifying studies

| | |
|---|---|
| Study selection criteria | ▶ Articles in English language published between 1 June 2008 and 31 October 2017.<br>▶ Studies conducted or reporting only on human subjects.<br>▶ Studies reporting on empirical, interpretive and critical research using any type of study methodology or study designs (case–control study, observational study, surveys, research reports and case reports).<br>▶ Studies reporting on any types of healthcare setting including primary care, hospitals, allied health practices or emergency departments. |
| Participant selection criteria | ▶ Studies conducted or reporting only on participants aged 60 years and above.<br>▶ There will be no limitation on upper age and gender of the participants.<br>▶ There will be no limitations on geographical location of the study participants. |
| Specific exclusion criteria | ▶ Studies reporting on non-human subjects.<br>▶ Studies not reporting on individuals aged 60 years and above.<br>▶ Studies reported in another language than English. |

that is, older people. The scoping review will be supplemented by a parallel study involving qualitative interviews with older people who have experienced frequent acute hospital presentations.

## RESEARCH METHODS
### Literature search
A pragmatic publication date cut-off point of June 2008 was adopted to focus on studies that were conducted after the publication of a working definition of integrated care by WHO.[16] The scoping review will use the six steps described by Arksey and O'Malley[23] for conducting a scoping review. These steps will be used as a guide to identify, select and review the literature. A Preferred Reporting Items for Systematic Reviews and Meta-Analyses (PRISMA) chart is attached as online supplementary appendix 1.

**Table 2** Summary of electronic search

| Database | Keywords used | No of publications identified | No of publications included |
|---|---|---|---|
| | | | |
| | | | |

**Table 3** Data extraction form

| Author and date |
|---|
| Title of the study |
| Aim of the study |
| Additional research questions/objectives |
| Study design characteristics |
| Participant characteristics |
| Outcomes reported |
| Most important finding |
| Other relevant findings |
| Conclusions |
| Study limitations |
| Authors recommendations |

### Step 1: identifying the research question
Integrated care is defined by the WHO as, '*The management and delivery of health services so that clients receive a continuum of preventive and curative services, according to their needs over time and across different levels of the health system*'.[24] This definition suggests that a wide array of healthcare activities—including, but not limited to, frequency of consultation, quality of consultation, location of medical services, interaction with healthcare providers and satisfaction with the care provided to follow-up plans—form integrated care. Therefore, an iterative approach will be adopted for refining the research questions. This approach will enable familiarisation with the current literature on the topic: to synthesise knowledge and information from relevant studies while helping to identify the knowledge gaps. The broad research question that we will start with is:

'What are the findings of research on older patient views, perspectives, expectations and experiences of integrated care?'

### Step 2: identifying the relevant studies
The literature search will be performed from August 2017 to December 2017, using PubMed, Web of Science, Embase, Google Scholar, Cochrane Library, CINAHL and ProQuest Dissertations. Terms specific to each individual database will be used for searching articles. Selection criteria defined in table 1 will be followed for inclusion or exclusion of articles in the scoping review.

### Step 3: study selection
Two experienced researchers will independently conduct an initial search of literature to identify relevant articles. A priori set of search terms will be formulated prior to conducting the initial search. A full list of these keywords will be provided in the follow-up publication. Articles will be selected by scanning their titles and abstracts. Abstracts of the identified articles will be reviewed collectively by both the researchers. On reviewing abstracts, potential studies will be identified. Full texts of the potential studies will be sourced and screened for the inclusion criteria. If

the text of potential studies is not available, full texts will be requested from the authors. A third reviewer will be included to help with final selection of the studies. All disagreements within the team will be resolved by mutual discussions. Published as well as grey literature will be searched for potential articles. Reference lists of included articles will also be hand-searched for potential studies. A PRISMA chart will be used to document the study selection procedure, and a summary of the electronic search will be recorded using the format described in table 2. We will use The Joanna Briggs Institute critical appraisal checklist[25] (qualitative research) and the Strengthening the Reporting of Observational Studies in Epidemiology statement[26] (quantitative research) to appraise the quality of final studies included in the scoping review.

### Step 4: data extraction

Data will be extracted from each included study using a systematic approach, and measures will be taken to maintain uniformity in the data extraction process. A standardised data extraction form will be developed and used for charting the data, adopting the titles described by Arksey and O'Malley.[23] Data extraction will be limited to and focused on the research question. The variables to be included for data extraction are as indicated in table 3.

### Step 5: reporting of the results

Synthesis of data is not typically a central objective for a scoping review. However, depending on the nature of the data collected, we will provide a narrative synthesis or apply qualitative (thematic analysis) and quantitative methods (descriptive statistics such as percentage) to describe the extent and nature of the studies included. Charts and tables will also be used to map the study findings and provide an overview of the concepts, main sources and types of evidence in the research area under review.

### Step 6: consulting

Levac and colleagues[27] suggest that consultation be adopted to provide insight and input beyond the literature. To ensure a person-centred approach, a local advocacy group working for older people will be engaged in the process of the scoping review. To facilitate wider knowledge translation activities, the scoping review findings will be disseminated among older patients', their families and other stakeholders.

### STUDY DISSEMINATION AND ETHICS

The scoping review findings will be published in a peer-reviewed journal, presented at public forums and conferences and will help determine the value of undertaking a full systematic review. The results from this scoping review will inform and guide the next phase of a multi-stage knowledge translation study. The review findings will be made accessible to providers, policy makers and consumers to make effective use of the findings.

### DISCUSSION

The study aims to review the literature on patients' perspectives, expectations and experiences of integrated care. This study aims to enhance our understanding of integrated care from an older person's perspective. As such, it addresses an urgent need for establishing person-centred care in the community setting in order to reduce the burden of fragmented care on patients', their informal and formal carers, health systems and society. We intend to define 'integrated care' from the patient perspective and to identify facilitators and barriers to person-centred integrated care. It is anticipated that the findings from this review will enable healthcare providers, researchers and policy makers to more effectively tailor integrated care suited to patient needs.

**Contributors** MMM conceived the idea, developed the research protocol and methods and drafted and edited the final manuscript. AM and GH helped to refine and develop the research question and study methods and made meaningful contributions to the drafting and editing of the manuscript. All authors approved the final manuscript submitted.

**Funding** This research received no specific grant from any funding agency in the public, commercial or not-for-profit sectors.

**Competing interests** None declared.

**Patient consent** Not required.

**Provenance and peer review** Not commissioned; externally peer reviewed.

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
