## [Reviewer comments · BMJ Open]

ARTICLE DETAILS

TITLE (PROVISIONAL)	What integrated care means from an older person's perspective? A scoping review protocol
AUTHORS	Mittinty, Manasi; Marshall, Amy; Harvey, Gillian

VERSION 1 – REVIEW

REVIEWER	Yves Couturier University of Sherbrooke, (Qc) Canada
REVIEW RETURNED	17-Oct-2017

GENERAL COMMENTS	Very usefull scoping. More details may be put on phase 4-5-6 of the scoping review. For instance, how will you concretely involve users in your work? I suggest to adress explicitly two questions 1) if the point of view of users is always usefull, they could have no ideas about topics a litle bit for from them clinical concerns. How will you cope with that? 2) Which benefits for services designs will you expect?
---

REVIEWER	Grant Russell Monash University Australia
REVIEW RETURNED	22-Oct-2017

GENERAL COMMENTS	Thank you for the opportunity to review this manuscript: What integrated care means from an older person's perspective? A scoping review protocol for BMJ Open The article describes a protocol for a scoping review as a means to review the scope and range of literature on older person's perspective of enablers and barriers of integrated care. It results from a broader knowledge translation study involving the design implementation and evaluation of a person-centred approach to integrated care for frail older people. I will confine my review to that of a general reader of your journal. The authors provide a succinct introduction prior to a standard description of a scoping review that follows Arksey and O'Malley's framework. The paper flows well, is tightly organised and well written. I did have a few concerns about the focus of the review. It seems extraordinarily broad in that it seeks to cover integration in all health care systems and across all domains of care. The complexity of the challenge is increased when considering a lack of clarity in the focus of the review (top of page 5) The search seems directed at a few different constructs
--

	 • Different models of integrated care • The “validation” of the patient’s perspective in these models • Enablers and barriers of integrated care from the patients’ perspectives (sic). And then contextualised barriers and enablers from the patients’ perspective. • Older people who are frequent users of potentially preventable acute care. • Frail older people In addition I feel that the justification as presented seems to see the benefits of integration being restricted to reducing unnecessary acute hospitalization. It does seem a little strange to limit to that outcome given the multiplicity of other potential consequences. Secondly, the authors could and should have better characterised the benefits of gaining a patient perspective. :Leaving it as a justification being that evidence based management of chronic disease blends patient expectation with clinical evidence seems insufficient. In addition I did note that there a few typographical errors in the document and hope that they can corrected in future iterations. All the above concerns are remediable, and I anticipate that the authors would have little difficulty in responding. However, after some thought I do not feel that the article warrants publication in its current form. My concern is that I don't feel that there is sufficient novelty in the question or the methodology to justify publication. To explain, there has been a recent trend for the publication of protocol papers. In general this has been a good thing. It allows for more detail to be added to a study design than would be permitted in a report of study findings and also allows novel methodologies to be disseminated and discussed. Both approaches add to knowledge. Reviewing the paper through a lens of novelty and contribution to the literature I can only really take away the fact that someone is seeking to do an extraordinarily broad scoping of the literature regarding integration using a standard and generic scoping protocol. I don't have problems with the research question (which although broad, is reasonable) or even its justification (which, as I have said does lack some focus). There is nothing wrong with the protocol, there just isn't enough in it for me to feel as a reader that it warrants publication. A few things would have helped the task of adding novelty or something substantial to the literature:  • A logic model of the conceptualization of the topic • A theoretical framework to inform the review (not critical but potentially valuable) • A diescription of the context of the study (the authors of just stated that there is an interest in South Australia that relates to the frail elderly) My advice would be that the authors give some thought to the argument and the underlying constructs underpinning the review, undertake and then submit a completed review. I would find it interesting and look forward to the completion of the work.
--	--

VERSION 1 – AUTHOR RESPONSE

Reply to Reviewer 1 comments:

1) How will you concretely involve users in your work? I suggest to address explicitly two questions

a. if the point of view of users is always useful, they could have no ideas about topics a little bit for from them clinical concerns. How will you cope with that?

b. Which benefits for services designs will you expect?

Author's response: We have addressed the reviewer's feedback into the Consulting section, page 9 which reads as follows:

"To ensure a person-centred approach, a local advocacy group working for older people will be engaged in the process of the scoping review. To facilitate wider knowledge translation activities, the scoping review findings will be disseminated among older patients, their families and other stakeholders."

Reply to Reviewer 2 comments

1) It seems extraordinarily broad in that it seeks to cover integration in all health care systems and across all domains of care.

Author's response: We thank the reviewer for their feedback. At this stage, we are going to start with a broad question focusing on integrated care for older people. Our initial searches suggest there is a limited amount of literature that specifically examines integrated care from the older person's perspective. However, if the broad scope of the review becomes an evident problem as we commence the review, we will re-visit this issue.

2) The authors could and should have better characterized the benefits of gaining a patient perspective.

Author's response: Addressing the reviewer's comments regarding justification of the benefits of gaining a patient-perspective of integrated, we note that this is currently a recognised gap in the literature, but one that empirical studies of integrated care initiatives have identified as important to address. We have addressed the gap in information in the background section.

We appreciate the comments from the reviewers. Thank you for reviewing our manuscript.

VERSION 2 – REVIEW

REVIEWER	Grant Russell Monash University
REVIEW RETURNED	03-Dec-2017

GENERAL COMMENTS	Thanks for sending the response to the review. In my original review I provided 6 suggestions / modifications, of which the authors responded, briefly, and a little tangentially, to two. I would have anticipated a little more reflection and a more in-depth response in what is an article that has broad similarity to its predecessor. As such I have little else to say beyond my original thoughts
---

VERSION 2 – AUTHOR RESPONSE

Reviewer comment

1. Breadth of the review

"I did have a few concerns about the focus of the review. It seems extraordinarily broad in that it seeks to cover integration in all health care systems and across all domains of care. The search seems directed at a few different constructs:

Different models of integrated care

The 'validation' of the patient's perspectives in these models

Enablers and barrier of integrated care from the patients' perspective (sic). And then contextualised barriers and enablers from the patients' perspective

Older people who are frequent users of potentially preventable care

Frail older people."

Authors' response: We acknowledge and agree with the reviewer that the scope of the review appears very broad. However, our initial scan of the literature suggests there is a limited number of paper that address issues relating to integrated care from a patient perspective. This will be our primary inclusion criteria. Papers that discuss the various issues relating to integration noted by the reviewer will only be included if they include data from a patient perspective. As noted, we expect there to be a small number of such papers. For this reason we are starting with a broad scope. If this becomes problematic as we commence the review, we will then re-visit our search strategy.

2. Justification of the benefits of integration

"..the justification as presented seems to see the benefits of integration being restricted to reducing unnecessary acute hospitalization. It does seem a little strange to limit to that outcome given the multiplicity of other potential consequences."

Authors' response: We have expanded the background and study rationale sections to provide a wider and more balanced reflection of the intended benefits of integrated care.

3. Benefits of gaining a patient perspective

"Secondly, the authors could and should have better characterised the benefits of gaining a patient perspective."

Authors' response: As noted above, we have expanded the background and study rationale sections to include more content around what is already known and not known about integrated care and the potential for differences between a medical and lay discourse. This is used to create the case for gaining a better understanding of patient perspectives of integrated care.

4. Typographical errors

"In addition, I did note that a few typographical errors in the document and hope that they can be corrected in future iterations."

Authors' response: We have undertaken a further edit of the manuscript to address the typographical errors.

5. Novelty and contribution

"A few things would have helped the task of adding novelty or something substantial to the literature:

A logic model of the conceptualization of the topic

A theoretical framework to inform the review (not critical but potentially valuable)

A description of the context of the study"

Authors' response: Thank you for this helpful comment. The scoping review was implicitly based on a conceptual framework known as the co-KT framework, which underpins the larger program of work, of which the scoping review forms one part. We acknowledge that we not had adequately described this in the earlier drafts of the protocol and have now provided a more detailed explanation of the larger program of work, the pilot study that has informed it and how the scoping review will contribute.

We appreciate the comments from the reviewers. Thank you for reviewing our manuscript.

VERSION 3 – REVIEW

REVIEWER	Grant Russell Monash University
REVIEW RETURNED	03-Jan-2018
GENERAL COMMENTS	Thanks for sending - the authors have addressed my original concerns with thought and perspective - I do think that the paper is now stronger and more suitable for publication in it present form.